# CT Features of Mallory–Weiss Syndrome

**DOI:** 10.3390/diagnostics15050623

**Published:** 2025-03-05

**Authors:** Romain L’Huillier, Adrien Patenotte, Alexandra Braillon

**Affiliations:** 1Department of Medical Imaging, Edouard Herriot Hospital, Hospices Civils de Lyon, University of Lyon, 69002 Lyon, France; 2LabTAU, INSERM U1032, 69003 Lyon, France; 3Everest, The French Comprehensive Liver Center, Hospices Civils de Lyon, University of Lyon, 69002 Lyon, France; 4Department of Gastro-Enterology, Edouard Herriot Hospital, Hospices Civils de Lyon, University of Lyon, 69002 Lyon, France; 5Department of Medical Imaging, Louis Pradel Hospital, Hospices Civils de Lyon, 69002 Lyon, France

**Keywords:** Mallory–Weiss syndrome, active bleeding, pneumatosis

## Abstract

We report in this clinical case Mallory–Weiss syndrome suspected on computed tomography (CT) and confirmed on endoscopy. Mallory–Weiss syndrome is a rare cause of upper gastrointestinal bleeding from vomiting-induced mucosal laceration(s) at the gastroesophageal junction. The description of Mallory–Weiss Syndrome is rare on imaging and this observation provides CT semiological elements useful in detecting signs of Mallory-Weiss syndrome.

**Figure 1 diagnostics-15-00623-f001:**
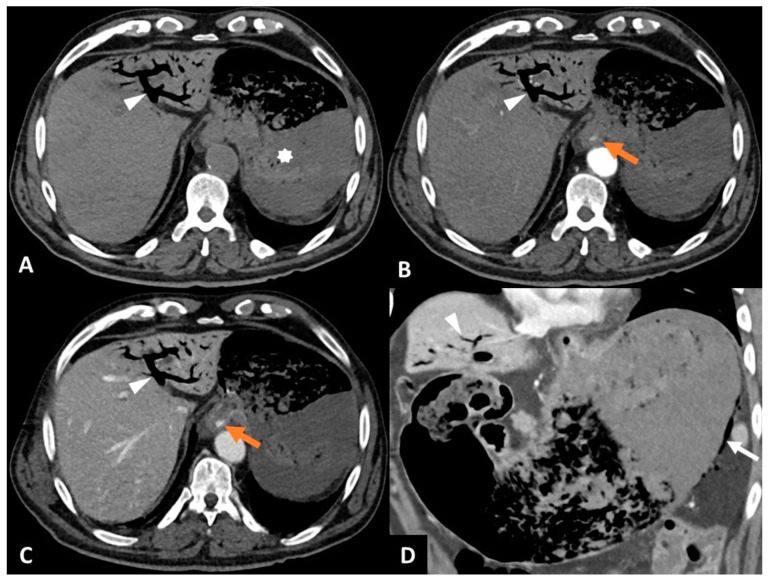
An 83-year-old man with Mallory–Weiss syndrome. An 83-year-old patient presented to the emergency department with a 12 h history of hematemesis and melena following an episode of vomiting. The patient had no history of alcohol consumption, gastroesophageal reflux disease, or diabetes. Upon clinical examination, there were tachycardia (130 bpm), dyspnea, normal blood pressure (122/82 mmHg), and acute abdominal pain. The hemoglobin level was normal (13.7 g/dL) initially, before decreasing (11.4 g/dL) two hours after admission. Given the loss of two hemoglobin points and the abdominal pain, with no immediate possibility of oeso-gastro-duodenal endoscopy, it was decided that an abdomino-pelvic CT should be performed. Conventional CT (SOMATOM^®^ Definition Edge, Siemens Healthineers, Erlangen, Germany) was performed before and after intravenous injection of iodinated contrast medium in the arterial and portal phases. Computed tomography revealed active endoluminal bleeding at the gastroesophageal junction, gastric parietal pneumatosis, and gas in the portal vein (Figure 1). (**A**) CT image in the axial plane obtained without intravenous contrast agent injection shows gas in the left portal branch (white arrowhead) and blood clotting in the stomach (asterisk). (**B**) CT image in the axial plane obtained during the arterial phase following intravenous injection of iodinated contrast agent shows endoluminal extravasation of contrast agent at the gastroesophageal junction (orange arrow) and gas in the left portal branch (white arrowhead). (**C**) CT image in the axial plane obtained during the portal phase following intravenous of iodinated contrast agent shows increased extravasation of contrast agent (orange arrow), indicating active bleeding at the gastroesophageal junction and gas in the left portal branch (white arrowhead). (**D**) CT image in an oblique coronal pane obtained during the portal phase following intravenous injection of iodinated contrast agent shows gas in the veins draining the greater gastric curvature (white arrow) and gas in the left portal branch (white arrowhead). There was no pneumoperitoneum or pneumomediastinum. Endoscopy confirmed active bleeding in the context of Mallory–Weiss syndrome (longitudinal mucosal laceration without transmural involvement at the gastroesophageal junction extending over 5 cm). Thermocoagulation of the bleeding and clipping were performed endoscopically. Treatment with intravenous proton-pump inhibitor was given for 3 days and oral proton-pump inhibitor therapy was initiated for 2 months. The patient was discharged 15 days after the endoscopy, with no recurrence of bleeding. Mallory–Weiss syndrome is a rare cause of upper gastrointestinal bleeding from vomiting-induced mucosal laceration(s) at the gastroesophageal junction [1,2]. During vomiting, the pylorus is closed and the intragastric pressure increases; the gastroesophageal junction is hyperextended, and its mucosa is lacerated [3]. The most commonly associated comorbidities are reflux esophagitis, diaphragmatic hernia, and alcohol abuse [1]. The description of Mallory–Weiss syndrome is rare on imaging and CT signs are sometimes difficult to detect gas in the esophageal wall [4] or parietal hematoma [5]. Unlike Boerhaave’s syndrome, in which all tunics of the esophageal wall are involved, there is usually no pleural effusion or extra digestive gas in the mediastinum [4]. CT can also rule out esophageal variceal rupture as a cause of gastrointestinal bleeding, given the absence of cirrhosis and signs of portal hypertension. In this case, the laceration probably reached a vein in the wall of the abdominal esophagus, explaining the diffusion of gas within the gastric wall and into the intrahepatic portal system. Diagnosis of Mallory–Weiss syndrome is based on oeso-gastric endoscopy, which reveals, in most cases, a single longitudinal mucosal laceration without transmural involvement at the gastroesophageal junction [2]. Endoscopy can also be used to treat active bleeding (argon plasma coagulation, thermocoagulation, epinephrine injection) and laceration (clipping) and to rule out other causes of upper gastrointestinal bleeding (peptic esophagitis, ruptured esophageal varices, etc.). The need for surgical therapy is exceptional [1]. Intravenous proton-pump inhibitors are always combined with local treatments. This observation provides CT semiological elements useful in detecting signs of Mallory–Weiss syndrome.

## Data Availability

The data presented in this study are available on request from the corresponding author. The data are not publicly available due to containing doctor and patient confidential information.

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
