# Peer review of "CT Features of Mallory–Weiss Syndrome"

_diagnostics, 2025, doi:10.3390/diagnostics15050623_

Round 1

Reviewer 1 Report

Comments and Suggestions for Authors

Overall evaluation:
This article describes a case of an 83 year old patient who presented with vomiting blood and black stool after vomiting. After imaging examination and endoscopic treatment, the final diagnosis was Mallory Weiss syndrome. The article provides a detailed imaging description, especially the presentation of CT images, and combines endoscopic examination results to demonstrate the rare imaging features of Mallory Weiss syndrome. The content of the article has clinical significance, especially for radiologists and gastroenterologists, providing valuable diagnostic references.
Specific opinions:
Clarity of case description:
The case description is relatively clear, with detailed records of the patient's chief complaint, medical history, physical examination, laboratory tests, and imaging results. Especially the description of CT images is very detailed, which helps readers understand the imaging manifestations.
Suggest adding more information about the patient's medical history in the case description (such as whether there is a long-term history of alcohol consumption, history of gastroesophageal reflux, etc.), as this information may have a significant impact on the occurrence of Mallory Weiss syndrome.
Accuracy of imaging description:
The imaging description is very detailed, especially the CT images, including portal vein gas accumulation, gastric wall gas accumulation, and active bleeding at the gastroesophageal junction. These descriptions help readers understand the imaging features of Mallory Weiss syndrome.
Suggest adding more details about CT scanning techniques, such as scanning parameters (layer thickness, reconstruction algorithm, etc.), in the imaging description, so that other researchers can reproduce similar imaging findings.
Rationality of diagnosis and treatment process:
The diagnostic process was reasonable, combined with imaging and endoscopic examinations, and the final diagnosis was Mallory Weiss syndrome. Endoscopic thermocoagulation hemostasis and clipping are standard treatment methods, and subsequent proton pump inhibitor therapy also complies with clinical guidelines.
Suggest adding more discussion on the diagnostic criteria and treatment guidelines for Mallory Weiss syndrome in the discussion section, so that readers can better understand the diagnosis and treatment process of this disease.
Depth of discussion section:
The discussion section is relatively brief, only briefly mentioning the rarity and imaging manifestations of Mallory Weiss syndrome. Suggest further expanding the discussion, including:
Epidemiology, risk factors, and pathophysiological mechanisms of Mallory Weiss syndrome.
Differential diagnosis between imaging findings and other causes of upper gastrointestinal bleeding, such as peptic ulcers, esophageal variceal rupture, etc.
Is the imaging manifestation of this case specific and can it provide reference for future diagnosis.
Reference citation:
The article cites relevant literature, but the quantity is relatively small. Suggest adding more literature references on the imaging manifestations and treatment of Mallory Weiss syndrome to support the argument of the article.

Author Response

Corrections are highlighted in yellow.

"Suggest adding more information about the patient's medical history in the case description (such as whether there is a long-term history of alcohol consumption, history of gastroesophageal reflux, etc.), as this information may have a significant impact on the occurrence of Mallory Weiss syndrome."

Response : added.

"Suggest adding more details about CT scanning techniques, such as scanning parameters (layer thickness, reconstruction algorithm, etc.), in the imaging description, so that other researchers can reproduce similar imaging findings."

Response: added.

"Suggest adding more discussion on the diagnostic criteria and treatment guidelines for Mallory Weiss syndrome in the discussion section, so that readers can better understand the diagnosis and treatment process of this disease."

Response: added.

"The discussion section is relatively brief, only briefly mentioning the rarity and imaging manifestations of Mallory Weiss syndrome. Suggest further expanding the discussion, including:
Epidemiology, risk factors, and pathophysiological mechanisms of Mallory Weiss syndrome.
Differential diagnosis between imaging findings and other causes of upper gastrointestinal bleeding, such as peptic ulcers, esophageal variceal rupture, etc."

Response: added.

"The article cites relevant literature, but the quantity is relatively small. Suggest adding more literature references on the imaging manifestations and treatment of Mallory Weiss syndrome to support the argument of the article."

Response: corrected.

Reviewer 2 Report

Comments and Suggestions for Authors

  1. Authors should briefly mention the pathophysiology of Mallory-Weiss syndrome to provide additional background for readers unfamiliar with this condition.
  2. Authors need to clarify the timeline of symptom progression and diagnostic steps for better reader comprehension. I believe that the authors should explain in the text why the patient with hematemesis and melena underwent CT first and only then EGDS. Did the patient have clinical signs of acute abdomen? 
  3. Authors need to ensure consistent terminology throughout the case (e.g., “gastroesophageal junction” vs. “oesogastric junction”).
  4. Authors need to consider expanding the reference list with additional recent studies on the imaging features of Mallory-Weiss syndrome.

Author Response

all corrections are highlighted in yellow. 

1 - Authors should briefly mention the pathophysiology of Mallory-Weiss syndrome to provide additional background for readers unfamiliar with this condition. 

Response 1: Added. 

2 - Authors need to clarify the timeline of symptom progression and diagnostic steps for better reader comprehension. I believe that the authors should explain in the text why the patient with hematemesis and melena underwent CT first and only then EGDS. Did the patient have clinical signs of acute abdomen? 

Response 2: Added. 

3 - Authors need to ensure consistent terminology throughout the case (e.g., “gastroesophageal junction” vs. “oesogastric junction”).

Response 3: Corrected.

4 - Authors need to consider expanding the reference list with additional recent studies on the imaging features of Mallory-Weiss syndrome.

Response 4: Added. 

Reviewer 3 Report

Comments and Suggestions for Authors

A Interesting Image has limited scientific value, but sometimes presents interresting findig, symptom, radiological image like in the case. 

The Authors could add to referrence list the case report by Zhi Hu and Dan Liu; NEJM 2024; 390:e49.

Author Response

"A Interesting Image has limited scientific value, but sometimes presents interresting findig, symptom, radiological image like in the case. 

The Authors could add to referrence list the case report by Zhi Hu and Dan Liu; NEJM 2024; 390:e49."

Answer : Thank you for the review.  Reference added.